

# Diurnal moths have larger hearing organs: evidence from comparative 3D morphometric study on geometrid moths

Pritha Dey[1,2], Max Söderholm[1] and Pasi Sihvonen[1]

[1] Finnish Museum of Natural History, Helsinki, Finland
[2] Current Affiliation: National Centre for Biological Sciences, Bangalore, India

## ABSTRACT

Tympanal organs, crucial for anti-bat defence in moths and key for taxonomy, are often overlooked due to their fragility during dissection. Using micro-CT, we analyzed the tympanal organs of 19 geometrid species, comparing diurnal and nocturnal species to understand how predators, like bats and diurnal birds or lizards, influence tympanal morphology and its allometric relationship with body size. We hypothesized that diurnal moths, with reduced anti-bat function, would have smaller tympanal organs, irrespective of body size. Allometry was tested using phylogenetic linear regression and tympanal volume was compared across diurnal and nocturnal moths relative to the abdominal volume. We used 3D geometric morphometry, followed by comparative analysis of the shape and size of ansa, a unique "mechanical" structure for geometrids. Contrary to our hypothesis, diurnal moths had significantly larger tympanal organs, with no allometric relationship with body size. Although activity patterns had no significant effect on ansa shape and size, convergence in shape among nocturnal species, suggests a possible auditory function. This study explores how daily activity patterns and predator-prey interactions may influence sensory adaptations, with larger tympanal organs of diurnal species potentially reflecting adaptations to detect lower, "non-bat" frequencies. It also highlights non-invasive imaging techniques for studying delicate anatomical features in museum specimens.

## INTRODUCTION

The animal world is either diurnal, nocturnal, or crepuscular; as the adaptations to function in one temporal activity pattern reduce the efficiency of functioning and survival in another (*Fraser, Metcalfe & Thorpe, 1997*). Nocturnality is presumed to be associated with adaptations to function in low-light and low-temperature conditions. However, in predominantly nocturnal taxa, like geckoes and moths, some clades have independently shifted to diurnal activity and evolved distinct adaptations suited to diurnal conditions like camouflage (*Fulgione et al., 2019*), photopic vision (*Sondhi et al., 2021*; *Kojima et al., In press*) aposematic (warning) wing colouration (*Fiedler & Brehm, 2021*) and hearing sensitivity to lower 'non-bat' frequency (*Fullard et al., 1997*). The shift to diurnality in these clades is likely driven by a combination of ecological factors such as predation,

Corresponding author
Pritha Dey, prithadey@ncbs.res.in

climate, and competition (*Gamble et al., 2015*). These factors can influence how animals send and receive information in their environment, which can lead to new selective pressures on sensory systems (*Sih, Ferrari & Harris, 2011*). For example, reduced bat predation during the day, prevalence of different visual predators, or changes in background sounds can all influence which sensory traits are most advantageous.

Predators are an important selective force in all terrestrial food webs. Prey species may invest proportionally more in defence, avoid interactions with their predators or follow alternative strategies for survival. For moths, the predation pressure from their most formidable predator, bats, vary in a predictable manner among nocturnal and diurnal species, influencing their daily activity pattern and hearing sensitivity (*Rydell, Jones & Waters, 1995*). This is reflected in the positive correlation between nocturnal flight activity and hearing sensitivity to ultrasound frequencies (*ter Hofstede, Ratcliffe & Fullard, 2008*). Shifting to diurnality is likely one of several strategies moths use to avoid bat predation.

Diurnal moths show marked degeneration of the ultrasonic hearing, and some of them are reported to be 'bat-deaf' (*Fullard et al., 1997*). There are reports of predator released moths with no ears, *e.g.*, the Polynesian Pyralid *Lathroteles obscura* (*Clarke, 1971*); and extremely poor sensitivity in female wingless geometrids (*Rydell et al., 1997*). While hearing sensitivity is often closely linked to body size, with larger animals generally having larger hearing organs and lower frequency hearing, as observed in lizards, frogs, mammals, and birds (*Hetherington, 1992*; *Werner & Igic, 2002*; *Gleich & Langemann, 2011*). Larger noctuid moths often have larger tympanal organs, but may be less sensitive to higher frequency bat calls compared to smaller moth species, like the crambid moths (*Fullard, 1988*; *Nakano & Mason, 2017*). Moreover, hearing sensitivity in tympanal insects depends on more than organ size-it is also influenced by the number of sensory neurons (*Loh et al., 2023*; *Göpfert & Wasserthal, 1999*), the mechanical properties of the tympanic membrane (*Lomas et al., 2011*; *Austin et al., 2024*), and external temperature (*Eberhard et al., 2014*; *Korsunovskaya & Zhantiev, 2007*). Since diurnal species are typically active during warmer periods of the day, thermal conditions may also affect their hearing performance. Therefore, the development and function of moth hearing organs, is not solely dictated by size, but likely reflect a complex interplay of physiological, ecological and environmental factors.

Hearing organs in insects have diverse functions, such as intraspecific communication, parasitic host localization, and predator avoidance (*Strauß & Lakes-Harlan, 2014*). One specialized form of these organs, the tympanal organs, typically consists of a thin membrane or tympanum backed by an air-filled sac, and has evolved independently in at least seven different insect orders (*Hoy & Robert, 1996*). In most tympanate insects, the highest sensitivity is between 15–80 kHz, i.e., within the bat calling frequencies (*Roeder & Treat, 1970*; *Kawahara & Barber, 2015*; *Hummel, Kössl & Nowotny, 2011*; *Miller & Surlykke, 2001*). This evolutionary pressure is particularly evident in nocturnal moths, where the multiple origins of hearing organs are thought to be a defense mechanism against insectivorous bats (*Roeder & Treat, 1970*; *Kawahara & Barber, 2015*; *Scoble, 1992*; *Ratcliffe & Fullard, 2005*; *ter Hofstede et al., 2013*). Most moths including Noctuoidea, Geometroidea and Pyraloidea possess ultrasound sensitive ears (*Fullard, 1998*). As such,
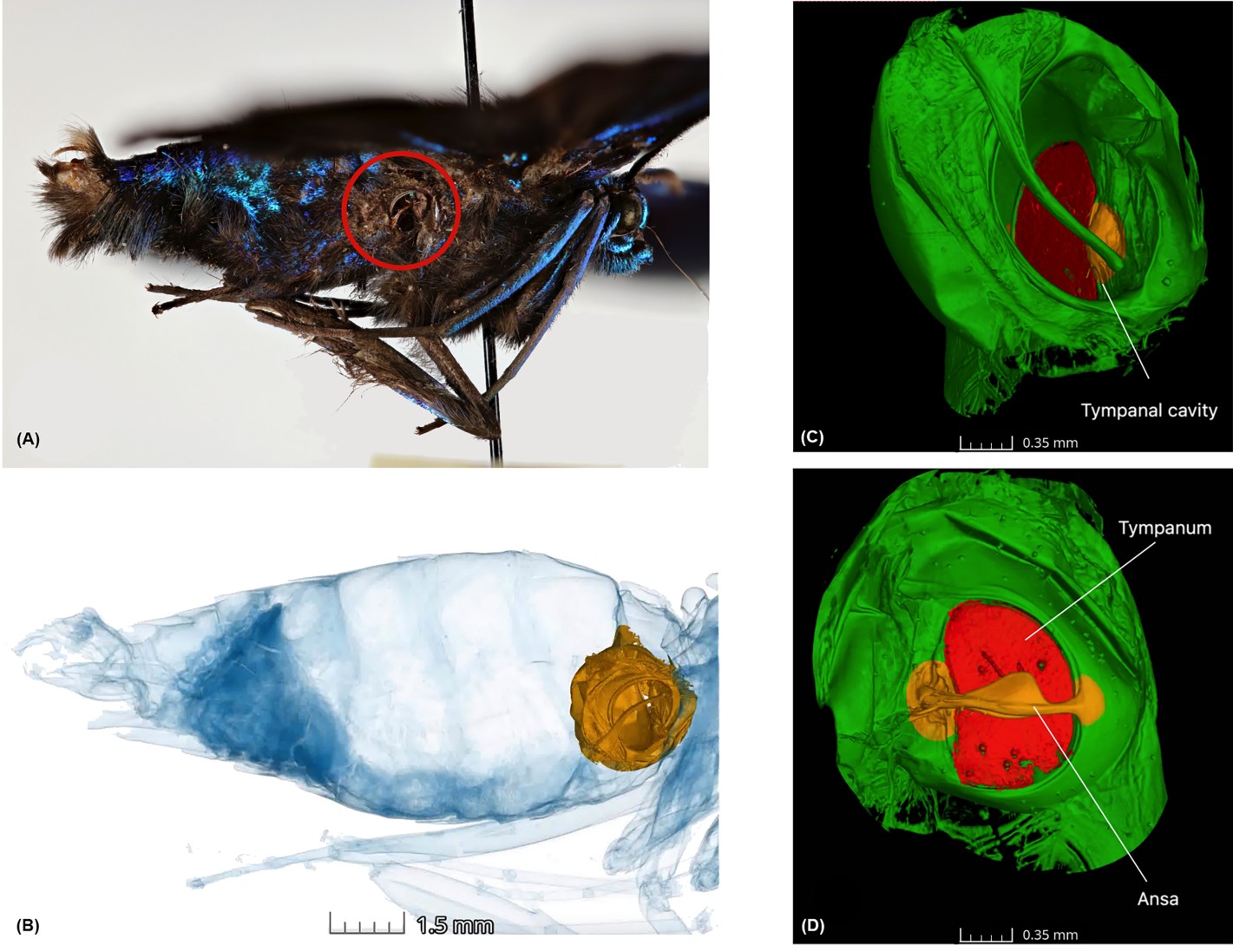

**Figure 1 Anatomy of the tympanal organs in geometrid moths.** The tympanal organ of the diurnal geometrid moth, *Milionia delicatul*a, a diurnal South-East Asian species in subfamily Ennominae where (A) shows the lateral side of the specimen, which was scanned, showing the opening of the abdominal tympanal organ (red circle); (B) shows the non-invasive micro-CT scan of the specimen clearly illustrating the tympanal organ (yellow) lodged in the abdominal cavity. All tissue structure other than the tympanal organ are in blue; (C) is the 3D reconstruction of the tympanal organ, showing the tympanal cavity on the outer side and (D) shows the ansa (yellow) on the inner wall of the tympanal cavity with the tympanic membrane or tympanum (red).

we would expect moths that are isolated from bats geographically or temporally to exhibit lower sensitivity to ultrasound (*Fullard et al., 1997*; *Surlykke, 1986*).

We chose geometrid moths as our model group due to well-documented diel activity patterns (*Lee et al., 2024*) and their use in testing hearing sensitivity in diurnal moths (*Surlykke et al., 1998*). While adult geometroid moths are primarily nocturnal, several unrelated lineages across families Uraniidae, Sematuridae, and Geometridae, exhibit diurnal activity (*Hausmann, 2001*; *Kawahara et al., 2018*; *Õunap, Viidalepp & Truuverk, 2016*). This transition has occurred independently in Lepidoptera multiple times

(*Sondhi et al., 2021*; *Kawahara et al., 2018*; *Kawahara & Breinholt, 2014*). The strictly diurnal geometrid moth *Archiearis parthenias*, which is temporally isolated from bats, is sensitive to around 12 kHz and practically deaf beyond 25 kHz (*Surlykke et al., 1998*), comparable to sympatric noctuid moths (*Rydell et al., 1997*; *Surlykke & Filskov, 1997*). The tympanal organ is an important taxonomic character for geometrids, with ansa being a prominent unique character of the family (*Cook & Scoble, 1992*; *Rajaei et al., 2022*) (Fig. 1C, and Video S1). The ansa, thought to protect the tympanum from mechanical damage caused by pressure from surrounding organs, particularly the oesophagus and flight muscles, has a wide base that strengthens the inner wall of the cavus, helping maintain tension across the tympanum (*Cook & Scoble, 1992*). The variation in ansa shape and size at the species and higher taxonomic level suggests it may serve functions beyond just a mechanical structure. However, research on diel activity in moths, function of hearing organ anatomy and predator pressures, remains limited.

In the above context, we wanted to ask, using a non-destructive 3D geometric morphometry: (1) Does the size of the tympanal organ show an allometric relationship with body size? (2) Is there a difference in the size of the tympanal organs between diurnal and nocturnal moths? (3) How do the shape and size of the ansa vary between diurnal and nocturnal species? The overarching question guiding this study is whether these differences are linked to specialized hearing adaptations for detecting predators. We hypothesize that the structure of the tympanal organs is driven more by diel activity patterns than by body size, with tympanal organs being smaller in diurnal moths due to reduced anti-bat function. Also, we hypothesize that the ansa plays a crucial role in auditory function—not just for mechanical strength—meaning its size and shape should vary depending on the moth's activity pattern.

## MATERIALS AND METHODS

We selected dried museum specimens of 19 geometrid species (seven diurnal, one cathemeral and 11 nocturnal). The study specimens with diurnal *vs* nocturnal species pairs, were chosen based on their relatedness classified in the same genus, same tribe, or same subfamily (*Murillo-Ramos et al., 2019*; *Brehm et al., 2022*; *Murillo-Ramos et al., 2023*). All study specimens are males, dry specimens deposited in museums, and they have been collected during the years 1993–2021, from different geographical locations, listed in Table S1.

The authors received permission from the National Biodiversity Authority, India to export and study specimens collected from India at the Finnish Museum of Natural History (Permit no: NBA/Tech Appl/9/INBA1202203315/22/22-23/132S).

### Micro-CT imaging and image processing

All the specimens were imaged using a Nikon XT H 225 micro-CT scanner. Specimens were positioned in the sample holder following the method described by *Moraes et al. (2023)* to minimize noise caused by the insect pin.

All imaging was performed using a molybdenum target. The first four specimens were imaged with the following parameters: 74 kV beam energy, 94 μA beam current, 500 ms

exposure time, 9,998 projections, and eight-frame averaging per projection. Each scan required 11 h. The remaining 15 specimens were imaged using adjusted settings: 80 kV beam energy, 84 µA beam current, 1.4 s exposure time, 4,476 projections, and four-frame averaging per projection, reducing the scan time to 7 h. Further testing showed that lowering the frame averaging to 2 did not significantly affect image quality, allowing the scan time to be further reduced to 3.5 h.

The voxel size of the reconstructed datasets ranged from 5 to 18 µm, with the voxels being isotropic. To enhance image quality and minimize deformation caused by desiccation, staining and critical point drying should be considered for future studies, although these methods require fresh specimens. Reconstructions were generated from the projection images using Nikon CT Pro 3D Version XT 6.9.1.

Segmentation was performed using VGSTUDIO MAX 2024.3. A spherical region of interest (ROI) was used for segmentation, as this approach facilitates the extraction of the tympanal organs due to their roughly spherical shape. When possible, excess material was manually removed from the ROI. The voxel-based 3D datasets were then converted into high-quality mesh models by first generating an isosurface model, customized for each dataset to achieve optimal results. To facilitate further analysis and manage file sizes, the final mesh models were reduced to 300,000 vertices and were exported as WRL and PLY files.

## Morphometric measurement and analyses

The body length was used as a proxy for overall body size in this study. In geometrid moths, the tympanal organs are located on the anterior side of the first and second abdominal segments. Given this anatomical placement, body length-which includes both the sclerotized thorax and abdomen-provides a more relevant and anatomically consistent measure of body size than forewing length, which is commonly used in lepidopteran studies (*Araújo Foerster et al., 2024*; *Brehm, Zeuss & Colwell, 2019*). Therefore, body length is a practical and biologically meaningful metric for our morphometric analyses. Using dried museum specimens may introduce some measurement bias, particularly due to potential shrinkage or deformation of the softer abdominal tissue during the drying process. We minimized the issue by excluding specimens with visibly distorted abdomens and applying consistent measurement protocols.

We used ImageJ software (*Schneider, Rasband & Eliceiri, 2012*) for this purpose. Further we measured abdomen width and abdomen length using Leica S9D stereo microscope, using Leica Flexacam C5 camera and Leica Microsystems's Enersight software v. 2024 (Leica, Wetzlar, Germany). For volumetric measurement of the tympanal organ, we used landmark based method on the 3D reconstructed model in the software AGMT-3D (*Herzlinger & Grosman, 2018*). We performed a phylogenetically informed linear regression to understand the relationship between body size and tympanal organ volume, while considering the relatedness of the species. To analyze the shape and size of the ansa, we used the *Geomorph* package in R (*Adams & Otárola-Castillo, 2013*), to do landmark-based 3D geometric morphometry on the 3D models of the tympanal organs. We digitized 10 homologous landmarks on the ansa of the tympanal organs of all the

species (Fig. S2). Then we performed a Generalized Procrustes Analysis, to remove the effects of scale on the landmarks, followed phylogenetic principal component analysis to compare and visualize the patterns the shape of ansa and compared the centroid size of the 3D model to compare the size of ansa, among the diurnal, cathemeral and nocturnal species. For the above analyses, we considered the relatedness of the species in our study using a phylogenetic tree constructed from mitochondrial cytochrome c oxidase subunit I (CO1) sequences. In this context, relatedness refers to genetic distance-that is, the evolutionary divergence between species as inferred from the differences in their CO1 DNA sequences. The models derive a covariance-matrix from the phylogenetic tree which reflects the expected similarity between species, which is then used to correct for phylogenetic non-independence. By using a phylogenetically informed framework, we made sure that similarities due to shared ancestry did not bias our results. We used Wilcoxon signed rank test, for pairwise comparisons in the analyses. We did not include the measurements for the cathemeral species *Dysphania percota* for any comparative analysis between diurnal and nocturnal taxa.

## RESULTS

The phylogenetic linear regression revealed no significant allometric relationship of tympanal organ volume with body size in geometrid moths (adjusted R-square = 0.06, $p > 0.05$) (Fig. 2A). This suggests that the size of the tympanal organ does not scale proportionally with the overall body size in this group. Notably, the diurnal moths were found to be significantly smaller in body size compared to their nocturnal counterparts (Fig. 2B). Despite the absence of a significant allometric relationship, we observed that diurnal moths possess significantly larger tympanal organs than nocturnal species ($p < 0.05$) (Fig. 3).

In terms of the ansa, no significant differences were found in the overall shape or size across taxa (Fig. 4). However, we did observe some evidence of convergence in the size of the ansa among the nocturnal species. The inter-quartile range for the ansa size in diurnal moths was 0.197, while for nocturnal moths it was 0.09 (Fig. 5). The micro-CT scans also reveal that the ansa is hollow with a narrow tube-like structure in the middle (Fig. 6 and Video S2). This indicates that while the ansa size does not vary significantly across all species, there may be a trend towards similar shape characteristics within nocturnal geometrid moths, possibly reflecting a shared evolutionary adaptation related to hearing sensitivity.

## DISCUSSION

In this study, we investigated how auditory structures in moths change as a function of body size, and the extent to which certain morphological traits predict hearing sensitivity. Our findings revealed several interesting patterns in the tympanal organ and ansa characteristics of geometrid moths. Despite no significant allometric relationship between tympanal organ volume and body size, we found that diurnal moths tend to have larger tympanal organs—despite being smaller overall—pointing to potential evolutionary adaptations that help them navigate different abiotic and biotic factors, such as avoiding

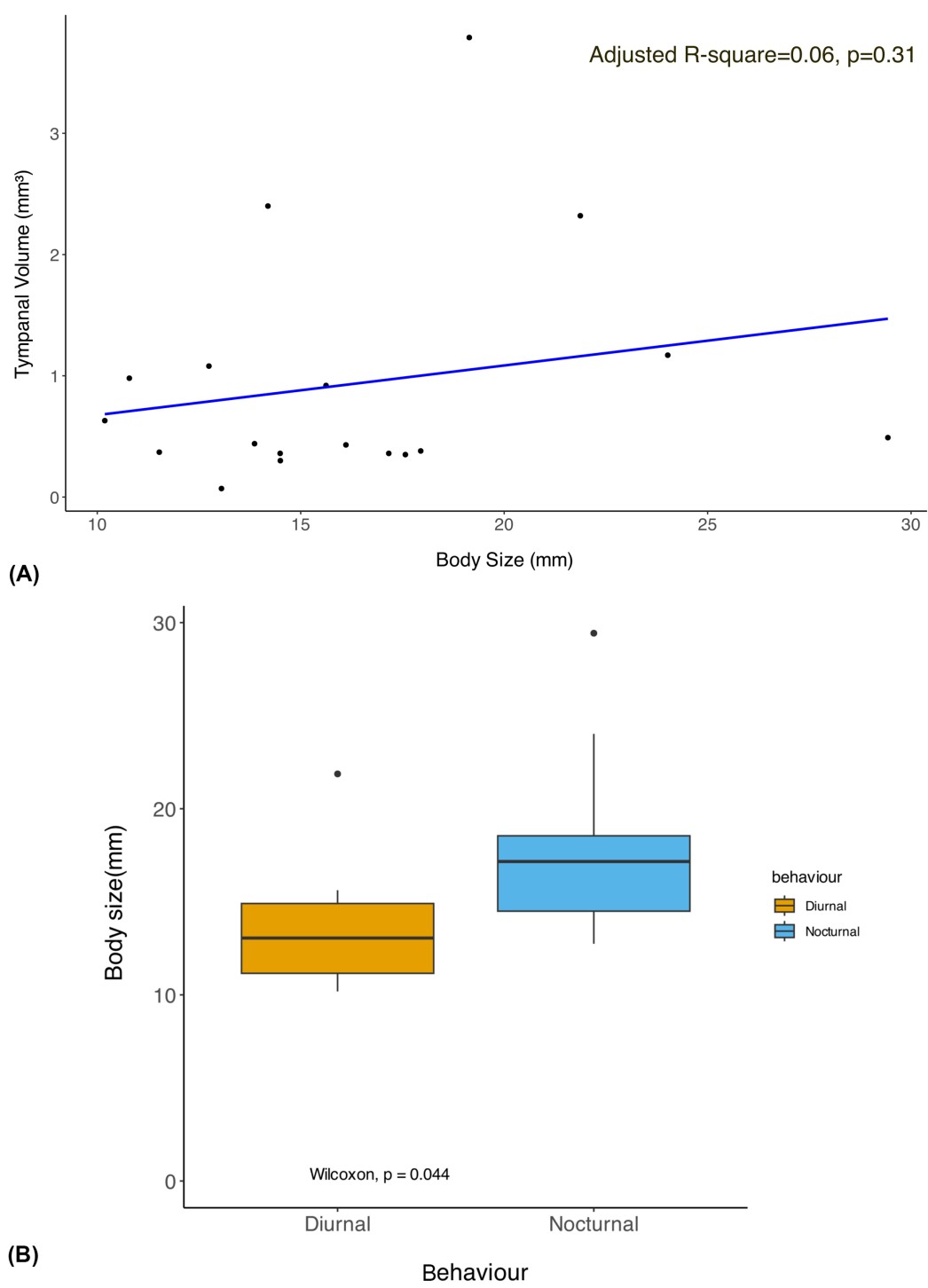

**Figure 2 Allometry of body size with tympanal organ size and comparison of body size among diurnal and nocturnal taxa.** (A) The plot shows the results of a phylogenetic linear regression, where the relationship between Body size and Tympanal volume is assessed while accounting for phylogenetic relatedness. Each data point corresponds to a species, and the regression line shows the best fit for the relationship between the traits depicting the non-significant ($p > 0.31$) allometric relationship; (B) the graph shows comparison of the body size of diurnal and nocturnal moths, with diurnal moths being significantly smaller than the nocturnal species ($p < 0.05$).

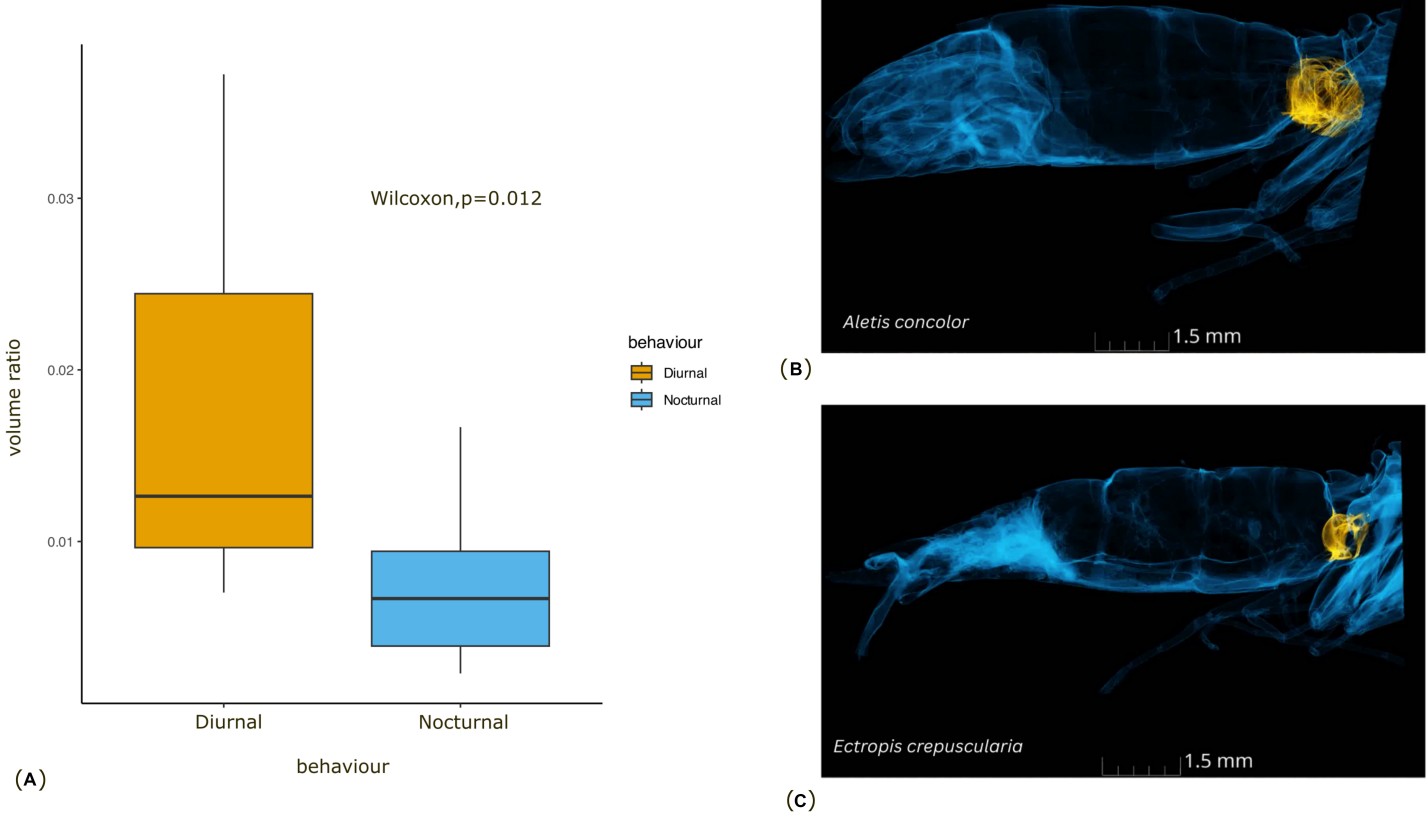

**Figure 3 Comparison of the tympanal organ size between diurnal and nocturnal moths.** (A) The graph shows the comparison of volume ratio (tympanal volume: abdomen volume) of the diurnal and nocturnal species. The diurnal species have significantly larger tympanal volume ($p < 0.012$) than the nocturnal species; (B) and (C) shows the micro-CT images of a diurnal (*Aletis concolor*) and a nocturnal (*Ectropis crepuscularia*) species, to visually represent the size of the tympanal organs (in yellow) compared to the abdomen size respectively. All tissue structure other than the tympanal organ are in blue.

diurnal predators. These include the flight sounds and calls of insectivorous birds, or the rustling vegetation sounds created by birds or other terrestrial predators like lizards. Such diverse acoustic cues may have shaped the tympanal organ morphology in diurnal moths (*Yack et al., 2020*).

In essence, our result means that the tympanal organ, originally assumed to function as an anti-bat hearing organ in nocturnal species, has adapted a new function in diurnal environment. The structure could still have a hearing function, potentially being adapted to different predator pressure on lower "non-bat" frequency and the larger size could serve more effectively this purpose.

Interestingly, the shape and size of the ansa do not differ significantly across taxa, though there is some evidence of convergence in the size of the ansa among nocturnal moth species. This suggests that the ansa may be playing a previously unexplored role in auditory function.

These findings highlight the complex evolutionary dynamics between predator-prey interactions and morphological adaptations in moths. Moths seem to be fine-tuning their

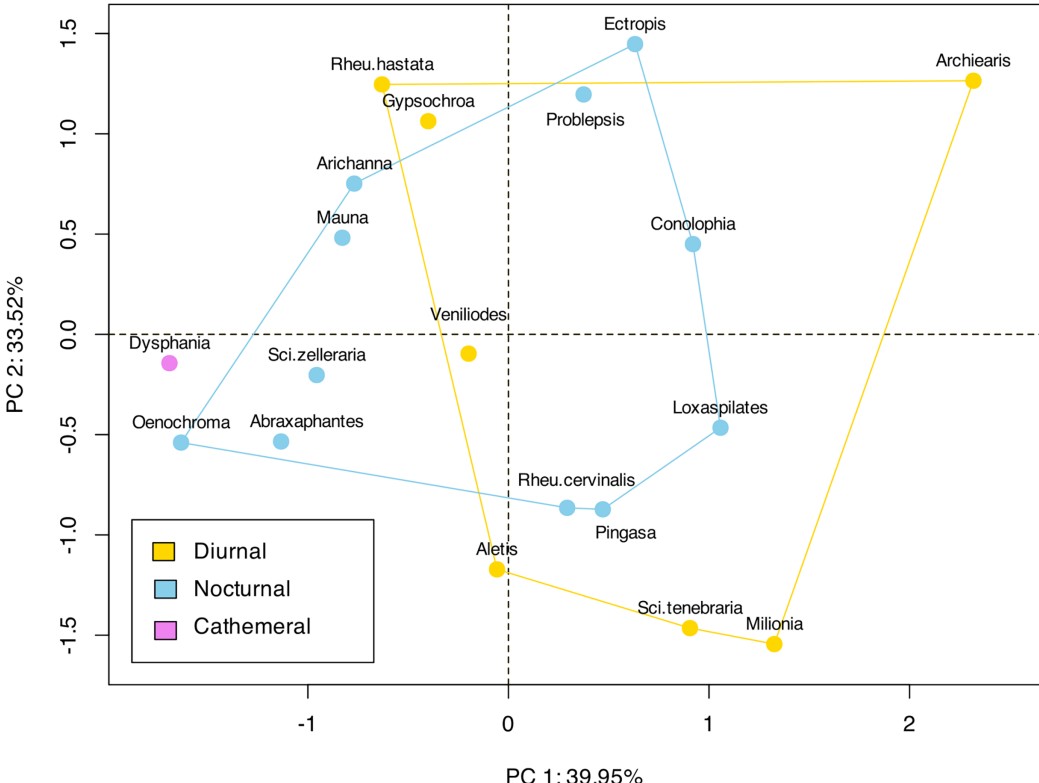

**Figure 4 Landmark based clustering of shape of ansa.** The plot shows the projection of the landmark-based shape parameters of each species onto the first two principal components (PC1 and PC2), which account for 39.95% and 33.52% of the total variance, respectively. Each point represents a species, and the colors indicate different categories or groups within the dataset. There is no significant clustering observed among the diurnal or nocturnal species.

auditory systems in response to their specific ecological niches, whether to optimizing hearing for bat evasion or adapting to the demands of diurnal activity. This research offers new perspectives on how sensory organs might contribute to survival and how predators could play a role in shaping highly specialized traits over time.

## Relationship between tympanal organ size and body size

Geometridae moths, which fly relatively slowly, rely more on their hearing based defence system to avoid bats, compared to other moth families, like Noctuidae, which fly fast, likely gaining some protection from predators independent of hearing (*Rydell et al., 1997*). The shift from nocturnal to diurnal behaviour has occurred independently multiple times across different subfamilies of geometrids, demonstrating parallel evolution. Diurnal species, though exposed to different environmental pressures, face similar predation threats, which may drive independent evolutionary trends such as the increase in the tympanal organ size, but decrease in body size. This phenomenon has been observed in Geometridae and other Lepidoptera families, where the transition from nocturnal to

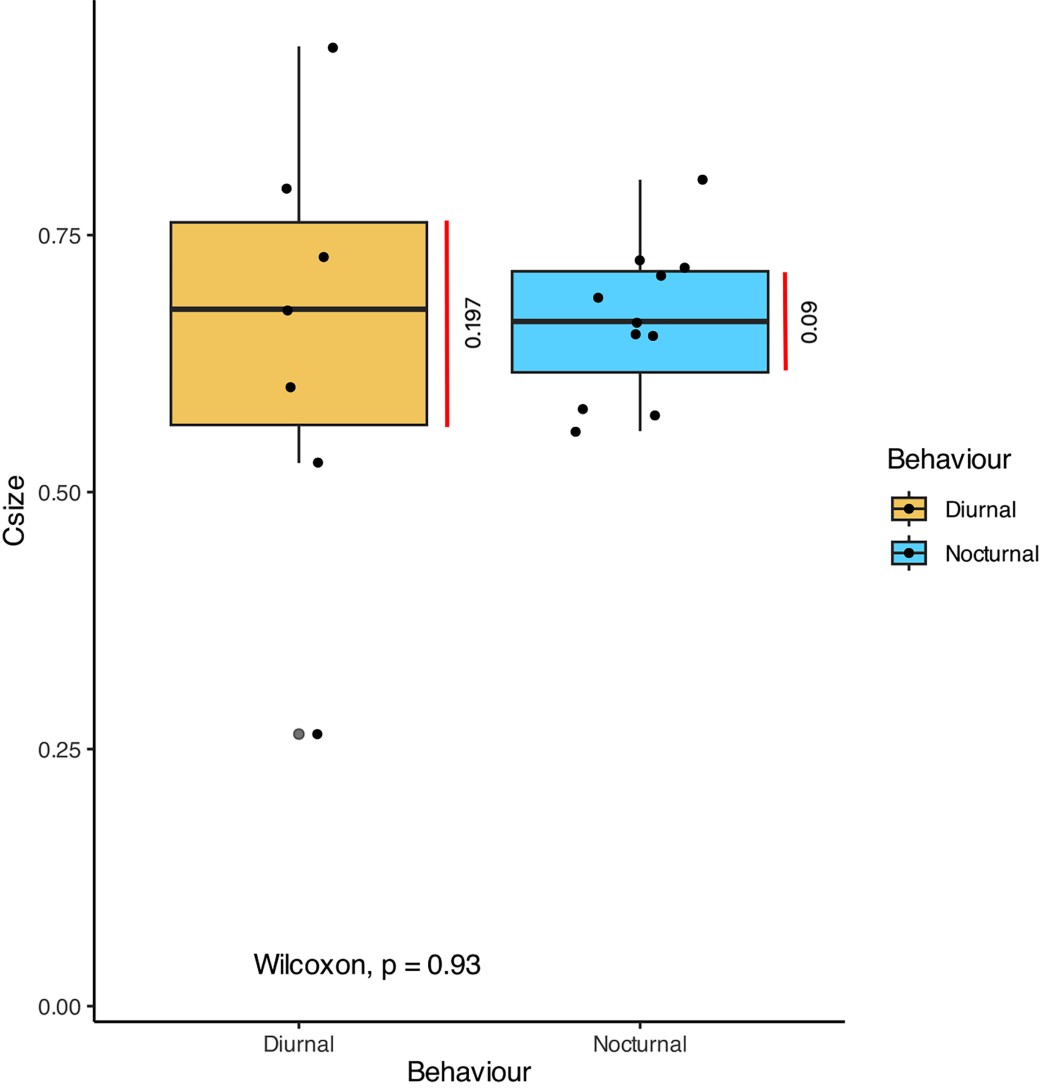

**Figure 5 Comparison of the size of ansa in diurnal and nocturnal moths.** The graph shows the comparison of Centroid size (Csize) of the 3D model of the ansa, which is an estimate of size of 3D model. There is no significant difference between the size of the ansa in diurnal and nocturnal taxa ($p > 0.93$), but the nocturnal species point towards a convergence in size (Inter-quartile range = 0.09).

diurnal flight suggests convergent adaptation to similar selective pressures (*Kawahara et al., 2018*; *Kawahara & Breinholt, 2014*; *Huemer et al., 2009*). Preliminary data from the Uraniidae family show that the diurnal species have visibly bigger tympanal organs (without morphometric analysis done) also (our observation, unpublished work). Smaller body sizes in day-flying moths have been observed in temperate areas, and those could be an adaptation to evade bird predation (*Tammaru et al., 2018*). This, however, needs to be tested in a wider geographical context. While the auditory systems of diurnal moths might be used for intraspecific communication or defense against birds, there is not yet sufficient evidence to confirm either of these suggestions.

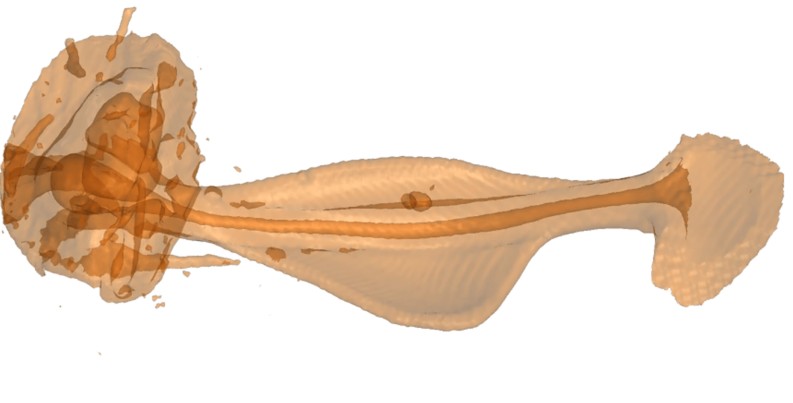

| | | | | |0.15 mm

**Figure 6 Structure of ansa.** Micro-CT scan of ansa of the tympanal organ of *Milionia delicatula*. The scan shows clearly that the structure is hollow (dark orange), being wider at the base, narrow in the middle, and slightly expanded on the apex. 3D video of the structure is available in Supplementary video.

## Tympanal organ size and hearing sensitivity

In anurans and lizards, larger tympanic membranes generally produce peak vibration amplitudes at lower frequencies than do smaller tympanic membranes (*Hetherington, 1992*; *Werner & Igic, 2002*) like how the area of a drum head affects its pitch (*Plassmann & Brändle, 1992*). Our approach in scaling of auditory structures and how their dimensions relate to hearing performance, gives valuable insights into how body size influences both the morphology of hearing organs and auditory capabilities. Similar relationships between tympanal size and hearing sensitivity have been observed in other insects as well (*Lehmann et al., 2010*), further supporting the idea that structural scaling can impact auditory function across taxa.

However, our findings suggest that this relationship may not be uniform across ecological contexts. There may be two opposing selective pressures at play: in nocturnal species, which tend to be larger, there could be a negative relationship between body size and tympanal organ size-potentially reflecting a functional limit beyond which further scaling does not enhance sensitivity to ultrasound. In contrast, diurnal species may show a more positive scaling relationship, possibly due to difference in acoustic environments or selective pressures not related to bat predation.

Additionally, structural modifications such as a thicker tympanum could represent alternative adaptations, potentially trading off sensitivity for increased durability in certain ecological scenarios. Different taxonomic groups may have developed unique adaptations to address size-dependent constraints on hearing. For example, larger moths may also rely on other defense mechanisms like increased flight speed or improved camouflage to evade bat predation (*Simon et al., 2023*). Examining deviations from proportional scaling between key auditory structures across insect species can reveal major evolutionary divergences and highlight instances of convergent evolution of the insect ear.

### Variation in the shape and size of the ansa

Auditory structures are shaped by a combination of phylogenetic, developmental, and physical constraints. For example, hearing ossicles in mammals and the columella in non-mammalian vertebrates are phylogenetic features that play well-established roles in auditory sensitivity. Developmentally, the growth and size of these structures are often linked, such as the relationship between the bulla and tympanic membrane in rodents (*van den Berge et al., 1990*). Physical constraints, such as the material properties of structures also play a role. In contrast, the ansa found in geometrid moths is known to support the tympanic membrane, but our study does not establish its contribution to hearing sensitivity. For understanding the evolution of auditory structures, it is essential to consider both ontogenetic and phylogenetic factors, as structural changes result from a balance of adaptation and inherent constraints, with ontogeny ultimately reflecting phylogenetic history.

This framework can be extended to the convergent evolution of auditory mechanisms in insects and mammals (*Montealegre-Z et al., 2012*). The ansa, only found in geometrid moths may function similarly to the middle ear ossicles in mammals, with variations in the ossicles influencing hearing sensitivity across mammalian species (*Hemilä, Nummela & Reuter, 1995*; *Nummela, 1995*). This comparison remains speculative, as the auditory role of the ansa has not been demonstrated. Similarly, the convergent shape and size of the ansa in nocturnal geometrid moths, along with its consistent presence across multiple species, suggest that it may serve a function beyond simple mechanical support. Although the ansa is a hollow structure-our micro-CT scans revealed a narrow tube-like internal feature that could potentially house a nerve or other tissue. This internal morphology, together with the ansa's close association with the tympanal membrane, raises the possibility that it contributes to the transmission of higher frequency vibrations. However, further investigation is needed to clarify the function of the ansa and its evolutionary implications.

To evaluate the adaptation *vs* constraints hypothesis, a comparative and functional analysis of various insect auditory systems is essential. Only by examining a broad range of species can we identify the shared acoustic properties as well as the functional differences between them, shedding light on the diversity of hearing systems in terms of size and frequency sensitivity.

## CONCLUSION

In conclusion, this study uses 3D geometric morphometry to hearing related insect cuticular structures in geometrid moths, offering new insights into the complex interplay of ecological pressures, evolutionary constraints, and morphological adaptations in this group. The lack of a significant allometric relationship between tympanal organ size and body size, alongside the presence of larger tympanal organs in diurnal moths, suggests that environmental factors, particularly predator-prey interactions, likely contribute to adaptations in auditory structures. The potential for convergent evolution, especially in the size and shape of the ansa across nocturnal species, further underscores the role of selective

pressures in shaping auditory morphology. We gain valuable insights into how body size influences both hearing capabilities and evolutionary trends. Ultimately, our study emphasizes the importance of considering environmental and phylogenetic constraints when interpreting the evolution of auditory structures and opens avenues for further investigation into the convergent mechanisms of hearing across taxa.

## ACKNOWLEDGEMENTS

We extend our sincere thanks to the members of the Forum Herbulot, a consortium of geometrid moth experts, for their valuable feedback, which significantly enhanced the study's conclusions and predictions.

### Funding

The work was supported by Research Council of Finland (decision # 331995, funding period 2020–2024). The funders had no role in study design, data collection and analysis, decision to publish, or preparation of the manuscript.

### Grant Disclosures

The following grant information was disclosed by the authors:
Research Council of Finland: # 331995.

### Competing Interests

The authors declare that they have no competing interests.

### Author Contributions

- Pritha Dey conceived and designed the experiments, performed the experiments, analyzed the data, prepared figures and/or tables, authored or reviewed drafts of the article, and approved the final draft.
- Max Söderholm performed the experiments, prepared figures and/or tables, authored or reviewed drafts of the article, micro CT and image analysis, and approved the final draft.
- Pasi Sihvonen conceived and designed the experiments, authored or reviewed drafts of the article, and approved the final draft.

### Data Availability

Data is available at:
Dey, Pritha; Sihvonen, Pasi; Soderholm, Max (2025). Data from: Diurnal moths have larger hearing organs: Evidence from comparative 3D morphometric study on geometrid moths [Dataset]. Dryad. https://doi.org/10.5061/dryad.prr4xgxz3

Morphosource: https://www.morphosource.org/projects/000731077?utf8=%E2%9C%93&per_page=100&locale=en

DOIs for individual MorphoSource entries are available in the Supplemental Files.

## Supplemental Information

Supplemental information for this article can be found online at http://dx.doi.org/10.7717/peerj.19834#supplemental-information.

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
