# Peer review of "Diurnal moths have larger hearing organs: evidence from comparative 3D morphometric study on geometrid moths"

_PeerJ, doi:10.7717/peerj.19834_

## Round 0.1 · original submission · Minor Revisions

Thank you for submitting your manuscript entitled " Diurnal moths have larger hearing organs: Evidence from comparative 3D morphometric study on geometrid moths." to PeerJ.

We have now received two reviews for your paper, which can be found attached to the end of this email. On the basis of these reviews and the Editor's own reading of the manuscript, I would like to see a major revision dealing with the comments. As you will see, both reviewers suggest areas where the paper could be improved, and we would like you to respond to these points in a revised version of your manuscript.

Reviewer 1 comments on the experimental design, particularly the interpretation of body length and the phylogenetic information used. He also questions some points in the conclusions, which appear to be highly speculative.

Reviewer 2 raises points that need to be clarified within the experimental design regarding some details of the biology of the species studied, which should be further detailed to justify their selection within the study. He also suggests refocusing the analyses by considering a possible relationship between the size of the tympanic organ and body size.

Please note that the reviewers' comments still need to be satisfactorily addressed before your paper can be accepted for publication.

·

Basic reporting

General comment: this is an interesting paper relating tympanum size in a moth taxon to nocturnal/ diurnal life style. Since bats, the most relevant predator group for nocturnal flying insects, are not active during day, diurnal moths obviously don’t “need” their ears for hearing bats. The hypothesis, that therefore they have a reduced tympanum has been falsified. The potential reasons are discussed.

The paper is well written and uses clear English.
Introduction and background present most relevant aspects. However, in statements that are more general one has the impression that the authors are much moth focused and are less informed about other hearing insects. I made specific comments in the pdf.
Structure is fine.
Figures are informative. Fig. 1a might be improved for visibility of the relevant structures. Legend to 3b,c might give more information about the colors. All lettering of the axes and within the figures appears to be too small to me.
Raw data – I didn’t see raw data beyond what is shown in the article.

Experimental design

Well defined. Description might be a bit more detailed (I made comments in the pdf) – for example why body length was chosen as proxy for size and what errors this might introduce. Also, what phylogenetic information exactly was used for the phylogenetically informed calculations.

Validity of the findings

In principle the findings are clear and sound; I only wonder whether body length is a good proxy for size in dried specimens. I made comments in the pdf. It needs to be written explicitly in M&M that the species studied are dried material. The inclusion of phylogenetic relatedness used for the correction of data needs to be explained in more detail – what data are used – is this genetic distance or some other value for phylogenetic distance or a more rough value like same genus, same tribe, same family, as one might have the impression when looking at the tables?
Conclusions:
The basic conclusions are justified, but several of the more extended assumptions appear to be too far reaching – and this is mostly so, since no physiological data are provided and also, since the function of the ansa is not understood yet – therefore assuming specific adaptation to hearing (like the ansa being hollow) are speculation. A comparison to hearing ossicles in tetrapods seems inappropriate to me.

·

Basic reporting

The manuscript "Diurnal moths have larger hearing organs: Evidence from comparative 3D morphometric study on geometrid moths" investigated whether anti-bat defense comes with larger tympanal organs, i.e., whether nocturnal moths have larger tympanal organs (in relative and absolute terms) than their diurnal cousins. By using one family of moths, Geometridae, where some of the subfamilies are diurnal, others nocturnal, provides an ideal model system as the location of the tympanal organ is within a cavity on the anterior side of the first abdominal segment (see fig 1 in "Simple ears – flexible behavior: Information processing in the moth auditory pathway") for all geometridae.
The introduction is concise, but not establishing what is meant by "more developed hearing organs" - not least as sensitivity is not a simple function of tympanal size. The number of neurons (scolophores), stiffness of the membrane but also humidity and temperature affect sensitivity - and diurnal species likely are active during warmer temperatures.

Experimental design

The study reconstructed the tympanal organ from 19 geometric species, but does not report the sex (antennae are dimorphic in some species) and age of the specimen, nor does table S1 report whether they are collected from tropical or temperate habitats. Is the choice of species solely by relatedness in the phylogenetic tree or also some convenience sampling? Biston betularia or Rhodometra sacraria (migratory) would allow to test some of the ecological questions. Does any of the 19 species emit ultrasound? Please note, I am not requesting more data, just more details on the choice by phylogenetic relatedness (see also comments on figure 2 below).
Insects can have really small ears (katytids), wondering what is the minimum size to be functional in geometridae? Are the hearing frequencies for all 19 species known? I assume not, but from the size of the tympanal sac and the ansa one might speculate the lower and upper frequency range.

Validity of the findings

Thank you for providing informative 3D reconstructions and videos.
The results are very interesting, as the assumed mechanism - function relationship (nocturnal species having larger ears) was not found. There might be two opposing forces, i.e., a negative relationship between tympanal organ size and body size in nocturnal species (no gain beyond a certain size, and since nocturnal species are larger, the tympanal organ has not scaled beyond what is needed to detect ultrasound) and a positive relationship in diurnal species. With so few species this might not reach statistical significance, but by color-drawing it in figure 2 and two regression lines, one for nocturnal species, one for diurnal species, one can visually inspect the allometric relationship. Alternatively, please provide more details on the phylogeny, i.e., tympanal size or volume ratio plotted against time of separation. If that is done in figure 2, the method section and result should be expanded, incl providing the raw data and equation as it is not obvious from the figure how phylogenetic relatedness is included in the body_size by tympanal_volume size graph. Minor: Please add to the graph that volume is in mm3 and body size in mm. Regarding figure 3, how would a separation into temperate vs tropical look like, i.e. is the volume ratio larger in temperate or tropical species?

---

## Round 0.2 · Minor Revisions

After reviewing this revised version of your manuscript, I see that the main comments suggested by the reviewers have been included. However, there are still some details that need to be clarified before having a final version that can be published. It is only necessary to clarify one point in the discussion concerning possible measurement errors in dry specimens.

·

Basic reporting

I thank the authors for the careful treatment of all pouints raised and the considerable changes made in the manuscript.

Experimental design

The only minor point left is that I do not find the "discussion of potential measurement errors" referring to the relatively soft abdomen in dried specimens, when body length was used as proxy for size - this additional discussion was announced in the rebuttal letter.

Validity of the findings

as before

Additional comments

none

·

Basic reporting

no comment

Experimental design

the authors have rectified the missing details

Validity of the findings

no comment

Additional comments

The authors have addressed all comments.

---

## Round 0.3 · accepted · Accept

After reviewing this revised version of your manuscript, I note that a paragraph has been included to clarify the point raised by the reviewer concerning possible measurement errors in dry specimens. Therefore, I am satisfied with the current version and consider it ready for publication.